# Effects of velocity based training vs. traditional 1RM percentage-based training on improving strength, jump, linear sprint and change of direction speed performance: A Systematic review with meta-analysis

Kai-Fang Liao[1,2], Xin-Xin Wang[1], Meng-Yuan Han[1], Lin-Long Li[1], George P. Nassis[3,4], Yong-Ming Li[1,5]*

1 School of Physical Education and Sport Training, Shanghai University of Sport, Shanghai, China, 2 Department of Strength and Conditioning, Guangdong Vocational Institute of Sport, Guangzhou, China, 3 Physical Education Department–College of Education (CEDU), United Arab Emirates University, Al Ain, Abu Dhabi, United Arab Emirates, 4 Department of Sports Science and Clinical Biomechanics, SDU Sport and Health Sciences Cluster (SHSC), University of Southern Denmark, Odense, Denmark, 5 China Institute of Sport Science, Beijing, China

* liyongming@sus.edu.cn

## Abstract

### Background

There has been a surge of interest on velocity-based training (VBT) in recent years. However, it remains unclear whether VBT is more effective in improving strength, jump, linear sprint and change of direction speed (CODs) than the traditional 1RM percentage-based training (PBT).

### Objectives

To compare the training effects in VBT vs. PBT upon strength, jump, linear sprint and CODs performance.

### Data sources

Web of science, PubMed and China National Knowledge Infrastructure (CNKI).

### Study eligibility criteria

The qualified studies for inclusion in the meta-analysis must have included a resistance training intervention that compared the effects of VBT and PBT on at least one measure of strength, jump, linear sprint and CODs with participants aged ≥16 yrs. and be written in English or Chinese.

**Data Availability Statement:** All relevant data are within the manuscript and its Supporting Information files.

**Funding:** This work was supported by the Ministry of Science and Technology of the People's Republic of China [2018YFF0300901] funds to Yongming Li. The funders had no role in study design, data collection and analysis, decision to publish, or preparation of the manuscript.

**Competing interests:** The authors have declared that no competing interests exist.

## Methods

The modified Pedro Scale was used to assess the risk of bias. Random-effects model was used to calculate the effects via the mean change and pre-SD (standard deviation). Mean difference (MD) or Standardized mean difference (SMD) was presented correspondently with 95% confidence interval (CI).

## Results

Six studies met the inclusion criteria including a total of 124 participants aged 16 to 30 yrs. The differences of training effects between VBT and PBT were not significant in back squat 1RM (MD = 3.03kg; 95%CI: -3.55, 9.61; $I^2$ = 0%) and load velocity 60%1RM (MD = 0.02m/s; 95%CI: -0.01,0.06; $I^2$ = 0%), jump (SMD = 0.27; 95%CI: -0.15,0.7; $I^2$ = 0%), linear sprint (MD = 0.01s; 95%CI: -0.06, 0.07; $I^2$ = 0%), and CODs (SMD = 0.49; 95%CI: -0.14, 1.07; $I^2$ = 0%).

## Conclusion

Both VBT and PBT can enhance strength, jump, linear sprint and CODs performance effectively without significant group difference.

## Introduction

Resistance training plays a pivotal role in enhancing strength, power, linear sprint and change of direction speed (CODs) performance in a wide range of healthy and athletic population [1–3]. It is well known that maximizing the resistance training effects largely depends on the manipulation of the program variables such as the intensity, volume, rest interval, duration, etc [4]. Among them, intensity and volume are the key variables in prescribing the program [4]. Traditionally, the resistance training prescription is based on percentage of one repetition maximum (%1RM) and repetitions, namely traditional 1RM percentage-based resistance training (PBT). Since its introduction by Thomas Delorme in 1940s [5], PBT has been studied and applied extensively, and proved to be an effective method by a huge body of researches. However, PBT has been criticized for the inherent limitations such as the complex process, risk of injury during the maximum strength test [6], and possible attenuation of type II muscle fiber adaptation owing to the sets to failure [7, 8], which may result in suboptimal training stimulus.

Over the past decade, there has been a surge of interest in using barbell velocity to measure and monitor the training intensity and volume, i.e. velocity based training (VBT), largely thanks to the major advancements in commercial velocity testing devices such as linear position transducers and accelerometers that allow for immediate feedback of repetition velocity [9]. VBT is a resistance training intervention that uses velocity feedback to prescribe and/or manipulate training load. Two new variables are adopted for prescribing the training load in VBT, one is the initial fastest repetition velocity in sets to set the load instead of %1RM, the other is the velocity loss threshold (VL) to terminate the set instead of the traditional fixed repetitions in sets. Compared with PBT, VBT has several compelling features. Firstly, velocity can be attained to estimate the 1RM and regulate the intensity in real time to match the actual intention for the particular session regardless of the fluctuation of personal 1RM due to the fatigue, nutrition, sleep etc [9]. Secondly, monitoring the velocity loss in training set can assist

to control the levels of efforts and fatigue in a certain range and make the lifted repetitions correspond well with the training specificity [6, 10]. Thirdly, instantaneous augmented feedback of velocity following each lifting repetition could motivate the athletes to enhance acute physical performance and improve adaptation accumulatively [11, 12]. Therefore, based on these advantages, it could be argued that the VBT would be more effective in improving physical performance than PBT. However, the answer to this question remains unclear so far.

Previous studies reported that both VBT and PBT could improve physical performance effectively [13–16]. More recently, some scattered researches have been carried out on comparing the training effects of VBT vs. PBT [17–21]. However, these researches to date have not been able to provide robust evidence for telling the distinctions of training effects on improving physical performance between VBT and PBT. For example, some studies confirmed that VBT improved strength more effectively than PBT due to the above-mentioned advantages [17], while the others demonstrated no significant differences between VBT and PBT [20, 22], and one study even reported a better effect in PBT [18]. Therefore, a systematic review with meta-analysis is warranted to draw a conclusion from the inconsistency.

The purpose of this study was to determine whether VBT was more effective than PBT in enhancing strength, jump, linear sprint and CODs. The hypothesis was that VBT would improve these outcomes better than PBT with a small effect. To the author's best knowledge, this is the first study to compare the effects of VBT vs. PBT by meta-analytic techniques. Our findings should enhance greater understanding about the different training effects and assist the practitioners to make better selection between VBT and PBT.

## 2 Methods

This systematic review and meta-analysis was conducted under the Preferred Reporting Items for Systematic Reviews and Meta-Analysis Protocol (PRISMA).

### 2.1 Search strategy

The first author searched the potential articles from PubMed, Web of Science and CNKI (China National Knowledge Infrastructure), up to January 6th, 2021. The following three English terms combined under Boolean syntax were applied in PubMed and Web of Science: term 1 "Velocity based training", VBT, velocity, speed, resistance; term 2 "Strength training", "power training", "resistance training", "percentage based training"; term 3 Strength, power, performance, "change of direction", agility, endurance, speed, sprint. Given the different language context, we adjusted the Chinese searching terms applied in CNKI as following: term 1: "基于速度的力量训练"(velocity based training), "速度负荷" (Velocity load); Term 2: 力量 (strength), 抗阻(resistance). The searching strategy and syntax are shown in S1 Table. Forward citation and reference lists of retrieved full texts articles were checked by hand searching to identify potential eligible studies that were not found by the initial search.

### 2.2 Eligibility criteria

Original research articles were eligible if they met the following eligibility criteria: 1) prospective randomized or non-randomized comparative trial; 2) implemented both VBT and PBT interventions; 3) the training interventions lasted at least 4 weeks; 4) the training frequency for both interventions was at least 2 times per week; 5) participants were healthy and aged ≥16 yrs.; 6) measures of muscle strength, jump, linear sprint or CODs performance was assessed before and after the intervention (minimum follow-up period of 4 weeks); 6) full-text was available in English or Chinese. VBT was defined as a resistance training intervention that uses velocity feedback to prescribe and/or manipulate training load. Movement velocity could be measured

with a linear position transducer, inertial sensor, force platform, and/or motion capture. PBT was defined as a resistance training intervention that prescribed training load based on a %1RM that was assessed at pre-intervention. We broadly defined resistance training as a sequence of dynamic strength exercises that utilized concentric and eccentric muscular contractions.

## 2.3 Outcomes

Outcomes were surrogate measures of physical performance. If the same study reported more than one outcome in the same outcome domain, we only extracted a single most relevant effect size to deal with the multiplicity according to a decision rule as following: 1) the most used test in included studies; 2) selected comparison with performance tests in accordance with resistance training practice (e.g. lower body performance > upper body performance; dominant leg > non-dominant leg); 3) selected estimates from randomized in preference to non-randomized comparison. Muscle strength outcomes included back squat or bench press 1RM tests, as well as barbell velocity outcomes included measures of load velocity profile (LVP) obtained at a specific relative load in the concentric phase of the back squat. Linear sprint included a timed maximal sprint between 5 and 30 meters in distance. CODs included a timed, pre-planned COD task characterized by a maximal sprint followed by a deceleration and re-acceleration to a new direction. We also included jump performance as an outcome. All outcomes were continuous measures.

## 2.4 Study selection

After the literature searches were complete, studies were collected into a single list using Endnote software (X9.3.3). The fourth author removed any duplicates, then the third and the fourth authors independently screened the titles and abstracts to exclude any unrelated articles, with the remaining full texts screened against the eligibility criteria by two authors (the second and the third author). Any conflict regarding selection of articles was resolved by consensus. Corresponding authors were contacted if a full-text manuscript could not be retrieved or to clarify aspects of the study in relation to the inclusion criteria.

## 2.5 Risk of bias assessment within individual studies

Two authors (the second and the fourth author) independently evaluated the risk of bias within included studies, with disagreements resolved by discussion and consensus. Kappa coefficient was applied to evaluate inter-raters' agreement. The result showed that the agreement was high (Kappa coefficient = 0.83).

Given lots of risk of bias assessment scales (Cochrane scale, Delphi scale, Pedro scale) are specialized for medical research, trials in sport science usually were evaluated as poor quality in accordance with these methodological scales [23]. We chose the scale (S2 Table) modified by Brughelli et al. [23] and Hooren et al. [24]. This scale is deemed more suitable for sport science research, and includes 10 items, with each item rated as: 0 = clearly no/not reported, 1 = maybe, and 2 = clearly yes. The articles were rated poor with a total score lower than 10, moderate with a score between 10 and 15, good with a score > 15, and excellent with a score equal to 20. Publication bias was evaluated by regression-based egger test of the intercept for small-study effect and visually inspecting a funnel plot.

## 2.6 Data extraction

All data were independently extracted by two authors (the third and the fourth author) to a data collection form in Excel (Microsoft Corporation, Redmond, Washington, USA). Any

discrepancy between two authors was resolved by discussion and consensus. Data items included: 1) participant characteristics, 2) sample size, 3) details of VBT and PBT interventions, 4) length of follow-up, 5) details of the outcome measure(s), 6) details of dropout rates, intervention adherence and adverse events, and 7) pre- and post-intervention data for each outcome measure (mean and SD). Study authors were contacted to obtain missing data wherever necessary.

## 2.7 Quality of evidence

The quality of evidence for each included outcome was rated using the evidence grading system developed by the Grades of Recommendation, Assessment, Development, and Evaluation (GRADE) collaboration. GRADE has four levels of evidence: very low, low, moderate and high. Pooled outcomes that included only randomized trials started with a 'high quality' rating, whereas pooled outcomes that included data from at least one non-randomized trial started with a low-quality rating. The evidence was then downgraded for each outcome based on the following domains: 1) risk of bias, 2) inconsistency of results, 3) indirectness of evidence, 4) imprecision of results, and 5) publication bias. The evidence was downgraded by one level if we judged that there was a serious limitation or by two levels if we judged there to be a very serious limitation. One authors (the first author) judged the quality of evidence. An overall GRADE quality rating was applied to the body of evidence by taking the lowest quality of evidence from all of the outcomes. Judgments about evidence quality were justified and documented within a GRADE evidence profile.

## 2.8 Statistical analysis

The review manager software (5.3) was applied for the meta-analytic. Random effects model for all outcomes was chosen to aggregate the effect size. $Chi^2$ and $I^2$ were calculated to test the heterogeneity. For $I^2$ values of 25, 50, and 75% represent low, medium, and high heterogeneity, respectively [25]. For $Chi^2$ with large value and $p < 0.1$ show evidences of heterogeneity. If $p > 0.1$ and $I^2 < 50\%$ provoked further investigation through a subgroup analysis of moderator variables (training experience, identity, gender, training frequency, training modalities, training weeks). In order to identify the presence of highly influential studies, a sensitivity analysis was executed by removing one study at a time. Studies were considered as influential if removal resulted in a change of heterogeneity ($p$) from significance ($p < 0.1$) to non-significance ($p > 0.1$).

If the same measures were applied in the same outcome, pooled mean difference (MD) was calculated with the intervention group's MD (mean difference from pre-test to post-test) and $SD_{pre}$. Otherwise, the SMD was calculated by dividing the MD by the pooled $SD_{pre}$ [26], the algorithm is as following:

$$[MD_{VBT} - MD_{PBT}/pooled\ SD_{pre}]$$

This algorithm was selected as it has been recommended for effect size calculation of independent pre-/post- study designs in meta-analysis based on simulation results. Both algorithms of MD and SMD in speed and COD outcomes were adjusted as $[M_{pre}—M_{post}]$, of which the smaller values represents better results compared with other outcomes. Statistical significance was set at $p < 0.05$. The absolute values of effect sizes were rated with the following criterion given by Cohen [27]: <0.2 as trivial, 0.2–0.49 as small, 0.5–0.79 as moderate, ≥0.8 as large. Values were reported with 95% confidence intervals to describe the range of the true effect. If the absolute value of aggregated effect and 95% confidence interval were above zero, effect size could be considered as clear evidence. A positive effect size indicated that the effect of VBT

was more effective in improving the performance than PBT, and a negative effect size indicated the opposite, i.e. PBT more effective than VBT. SMDs were also applied to do the funnel plot so that all estimates can be put into one plot.

## 3 Results

### 3.1 Search results

The initial search resulted in 6533 records in English and Chinese. After excluding 2485 duplicates, 4048 records screening by the title and abstract, 3987 papers were subsequently excluded. The remaining 61 articles were read in full-text for eligibility. At last, 6 studies were included in the qualitative and quantitative analysis. These data are presented in Fig 1.

### 3.2 Risk of bias assessment

According to the modified scale, the scores of the 6 included studies ranged from 10 to 17, the overall quality was moderate, 3 studies were rated as good [17, 20, 28], and 3 studies were rated as moderate [18, 22, 29]. All studies got high scores in item 5, 7 and 9, but no study adopted a control group (see Table 1). The funnel plot show that the effects were symmetric distributed around the overall pooled effect size (see Fig 2). Egger's test of the intercept indicated that there was no small-study effect (β = -0.24; 95%CI -3.67 to 5.8; $p$ = 0.64).

### 3.3 Studies' characteristics

The characteristics of 6 studies included in qualitative analysis were displayed in Table 2. A total of 124 subjects were included in the research, with age ranging from 16 to 30 yrs. The intervention duration ranged from 6 to 8 weeks with a training frequency of 2 to 3 sessions per

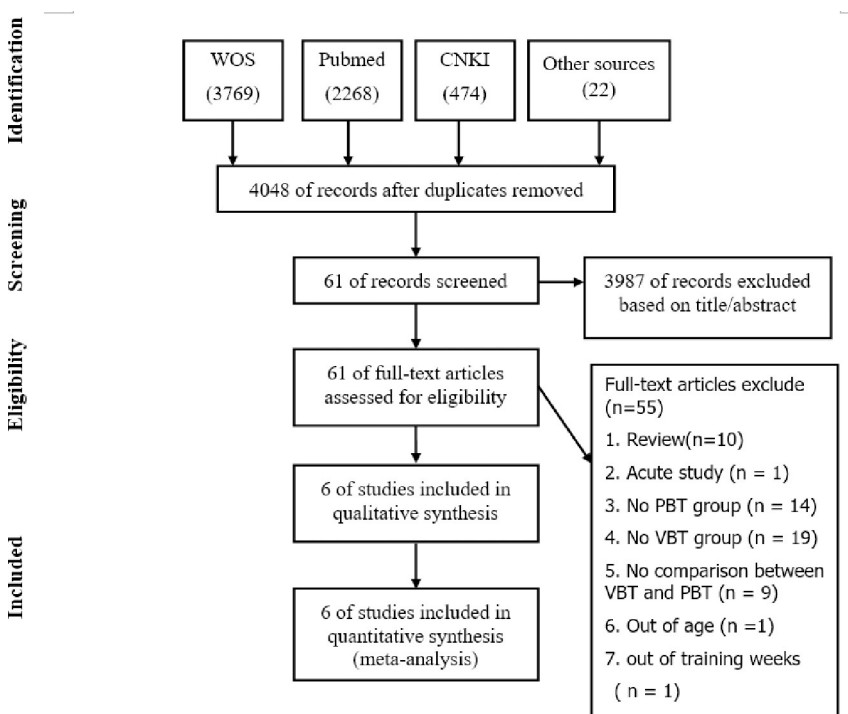

**Fig 1. Flow chart illustrating the different phases of the search and study selection.** WOS, web of science; CNKI, China National Knowledge Infrastructure; VBT, velocity-based training; PBT, Percentage-based training.

**Table 1. The results of risk of bias assessment.**

| Studies | Items | | | | | | | | | | |
|---|---|---|---|---|---|---|---|---|---|---|---|
| | 1 | 2 | 3 | 4 | 5 | 6 | 7 | 8 | 9 | 10 | Score |
| Chen Song et al., | 1 | 1 | 0 | 2 | 2 | 0 | 2 | 0 | 2 | 0 | 10 |
| Wang Zhihui et al., | 2 | 2 | 2 | 2 | 2 | 0 | 2 | 2 | 2 | 1 | 17 |
| Dorrell et al., | 2 | 2 | 2 | 2 | 2 | 0 | 2 | 1 | 2 | 1 | 16 |
| Orange et al., | 2 | 2 | 2 | 1 | 2 | 0 | 2 | 0 | 2 | 1 | 14 |
| Banyard et al., | 2 | 2 | 0 | 2 | 2 | 0 | 2 | 1 | 2 | 2 | 15 |
| Held et al., | 2 | 2 | 2 | 2 | 2 | 0 | 2 | 1 | 2 | 1 | 16 |

week. Among which, only 1 study had 3 sessions per week [22]. All studies took back squat as the intervention exercise, 2 studies also adopted other exercises such as bench press and dead-lift [17, 20]. The intensity of the resistance training ranged from 43% to 95%1RM or corre-spondent velocity. The sets were equal in both VBT and PBT groups ranging from 3 to 10. Four studies had the same repetitions in both groups ranging from 2 to 10, while 2 studies used the different repetition scheme, among which, 1 study adopted velocity loss [17] and 1 study used both fixed repetition and velocity loss as the termination of the sets [20]. The out-come measures included tests of maximum strength, power, barbell velocity, force, maximal oxygen uptake ($VO_2$max), linear sprint and CODs. All studies tested the back squat 1RM, while 4 studies tested countermovement jump (CMJ) tests [18, 20, 22, 28], and 3 studies tested speed [18, 22, 28]. In addition, 2 studies tested CODs [22, 28] and load velocity [18, 22], and 1

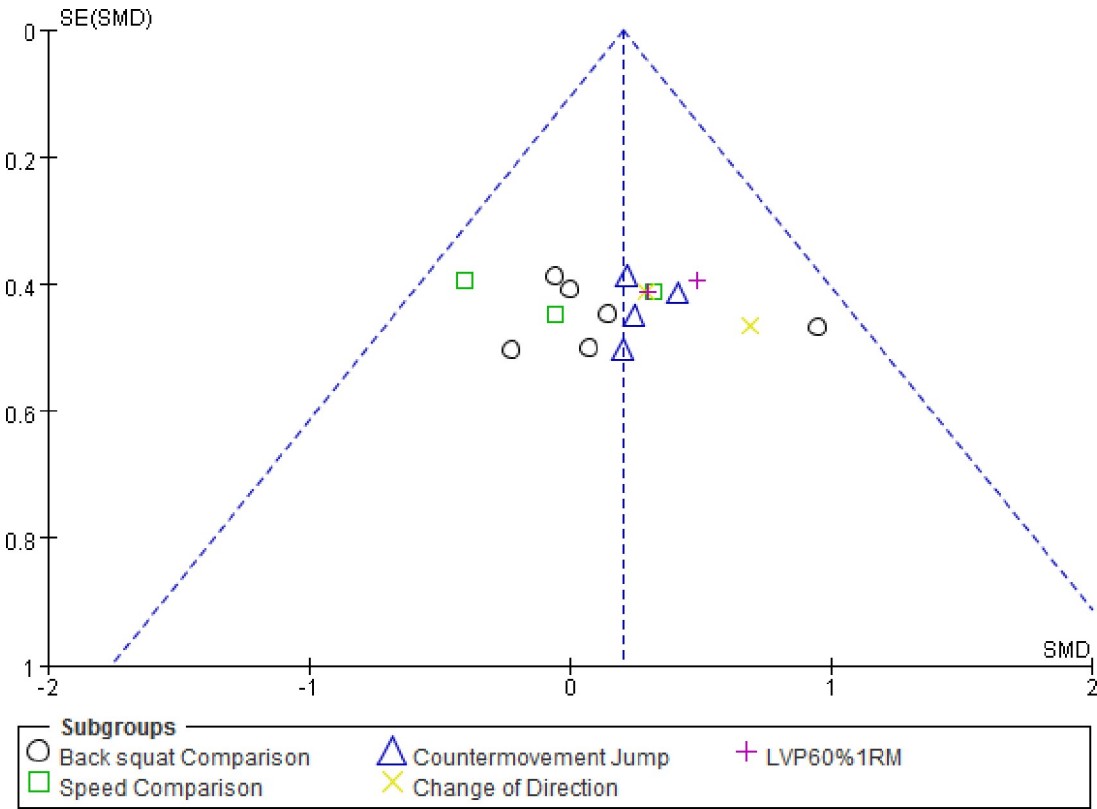

**Fig 2. The funnel plots.** LVP, load velocity profile; RM, repetition of maximum.

**Table 2. The characteristics of the included studies.**

| Study | Study Design | Participants | | | | | | Frequency (times/week) | Weeks | Interventions | | | Intensity (% 1RM) | Outcome Measure |
|---|---|---|---|---|---|---|---|---|---|---|---|---|---|---|
| | | Identity | Training Experience (years) | Age (years) | Sex | N | Drop out rate | | | Sets | Exercises | Reps | | Performance test |
| Orange 2019 | Randomized | Rugby players | ≥2 | 17±1 | M | 27 | VBT: 25% PBT: 6.25% | 2 | 7 | 4 | Back squat | 5 | VBT: velocity at 60–80% 1RM PBT:60–80% RM | **Back squat 1RM CMJ, Drop jump LVP-40%,*60%,80%,90%1RM **10,20,30m sprint *Sessional mean velocity *Sessional mean power *sessional TUT *sessional RPE |
| Banyard 2020 | Non-randomized | Resistance-trained | ≥2 | 25.5±5 | M | 24 | 0% | 3 | 6 | 5 | Back squat | 5 | VBT: velocity at 59.9–69.4% 1RM PBT: 59–85% | Back squat 1RM *PV-CMJ LVP-20%,40%,60%,80%,90%,100%1RM *5, *10, *20m sprint 15m shuttle *Sessional mean velocity *Sessional mean power *Sessional TUT *Sessional RPE |
| Song Chen1995 | Non-randomized | College athletes | -- | 18–22 | F | 20 | VBT: 20% PBT: 20% | 2 | 8 | 4 | Back squat | VBT:6–10 PBT:6–9 | VBT: Velocity at Maximum power load PBT:60–75% 1RM | Back squat 1RM Maximum Force Peak Velocity Peak power |
| Zhihui Wang 2020 | Randomized | Basketball players | 3.20±0.42 | 20.1 ±0.88 | M | 20 | 0% | | 2 | 4 | Back squat | 6 | VBT: velocity at 75% 1RM PBT:75% 1RM | Back squat 1RM CMJ 10 yard sprint *T test |
| Dorrell 2019 | Randomized | Resistance-trained | ≥2 | 22.8 ±4.5 | M | 16 | Total: 46.7% | 2 | 6 | 3 | Back squat Bench press OHP Deadlift | 2–8 2–8 2–8 2–8 | VBT: Velocity at 70–95%1RM PBT: 70–95% 1RM | Back squat 1RM *Bench Press 1RM OHP 1RM Deadlift 1RM *CMJ *Total volume |
| Held 2021 | Randomized | Rowers | ≥2 | 19.6 ±2.1 | F M | 4 17 | VBT:45.5% PBT:60% | 2 | 8 | 4 | Power clean Back squat Bench Row Deadlift Bench press | VBT:10%VL PBT: Sets to failure | VBT: velocity at 80% 1RM PBT: 80% | *Back squat 1RM *Bench Row 1RM Deadlift 1RM *Bench press 1RM VO$_2$MAX PVO$_2$MAX *Total repetitions *Change of overall stress *Change of overall recovery |

F, female; M, male; Reps, repetitions; PBT, percentage-based training; VBT, velocity-based training; N, number of participants; RM, repetition maximum; OHP, Overhead press; LVP, load velocity profile; TUT, time under tension; RPE, rating of perceived exertion;

*, in favor of VBT;

**, in favor of PBT

study tested the $VO_2max$ [17]. Four studies reported the sessional measures such as total volume, RPE, time under tension, barbell mean velocity and mean power [17, 18, 20, 22].

## 3.4 Quantitative analysis

The performance tests in the included studies could be categorized into 4 kinds after dealing with the multiplicity (Table 3), including strength consisted of back squat 1RM and load velocity with 60%1RM, CMJ, linear sprint and CODs. The evidences quality was low. These data were showed in Table 4.

**3.4.1 Strength.** *1) Back squat 1RM*. The pooled results from 6 studies consisting of 124 participants suggested that there was no significant difference in improving back squat strength between VBT and PBT ($p = 0.37$). The heterogeneity was low ($Chi^2 = 4.28$, $p = 0.51$; $I^2 = 0\%$). In random effects model, aggregated MD and 95%CI were 3.03kg (-3.55, 9.61) in favor of VBT. These data were showed in Fig 3A.

*2) Back squat LVP with 60%1RM*. LVP with 60%1RM was chosen because 1m/s (~60% 1RM) was usually used to evaluate the performance in resistance training [22, 30]. The pooled results from 2 studies consisting of 51 participants indicated that there was no significant

**Table 3. Performance indicators of included studies.**

| Study | Outcome | Performance indicators | Selected for meta-analysis |
|---|---|---|---|
| Dorrell et al., | strength | Back squat 1RM | √ |
| | | | |
| | | Bench press 1RM | |
| | | Deadlift 1RM | |
| | | Overhead press 1RM | |
| Held et al., | Strength | Back squat 1RM | √ |
| | | Bench press 1RM | |
| | | Deadlift 1RM | |
| | | Bench row 1RM | |
| Orange et al., | Jump | Countermovement jump | √ |
| | | Squat jump | |
| | Speed | 10m | √ |
| | | 20m | |
| | | 30m | |
| | Load velocity | 40%1RM | √ |
| | | 60%1RM | |
| | | 80%1RM | |
| | | 90%1RM | |
| Banyard et al., | Speed | 5m | |
| | | 10m | √ |
| | | 20m | |
| | Load velocity | 20%1RM | |
| | | 40%1RM | |
| | | 60%1RM | √ |
| | | 80%1RM | |
| | | 90%1RM | |
| | | 100%1RM | |
| | Change of direction | Dominant leg 15m shuttle | √ |
| | | Non-dominant leg 15m shuttle | |

**Table 4. GRADE of evidence profile.**

| | Summary of findings | | | Quality assessment | | | | | |
|---|---|---|---|---|---|---|---|---|---|
| Outcome | No. of participants (studies) | Pooled effects (95%CI) | $I^2$ | Risk of bias | Inconsistence | Indirectness | Imprecision | Publication bias | Quality rating |
| Back squat (kg) | 124(6) | 3.03(-3.55, 9.61) | 0% | Serious limitation[a] | No serious inconsistence | No serious in indirectness | No serious imprecision | Undetected | Low |
| LVP60%1RM (m/s) | 51(2) | 0.03 (-0.01, 0.07) | 0% | Serious limitation[a] | No serious inconsistence | No serious in indirectness | No serious imprecision | Undetected | Low |
| CMJ | 87(4) | 0.27(-0.15, 0.70) | 0% | Serious limitation[a] | No serious inconsistence | No serious in indirectness | No serious imprecision | Undetected | Low |
| Linear sprint (s) | 71(3) | 0.01 (-0.06, 0.07) | 0% | Serious limitation[a] | No serious inconsistence | No serious in indirectness | No serious imprecision | Undetected | Low |
| CODs | 44(2) | 0.46(-0.14, 1.07) | 0% | Serious limitation[a] | No serious inconsistence | No serious in indirectness | No serious imprecision | Undetected | Low |

a, one studies included in the outcome was non-randomized design; LVP60%1RM, load velocity profile with 60% one repetition maximum; CMJ, countermovement jump; CODs, change of direction speed

difference in improving LVP60%1RM between VBT and PBT ($p$ = 0.21). The heterogeneity was low (Chi$^2$ = 0.96, $p$ = 0.33; $I^2$ = 0%). In random effects model, aggregated MD and 95%CI were 0.02m/s (-0.01, 0.06) in favor of VBT. These data were showed in Fig 3B.

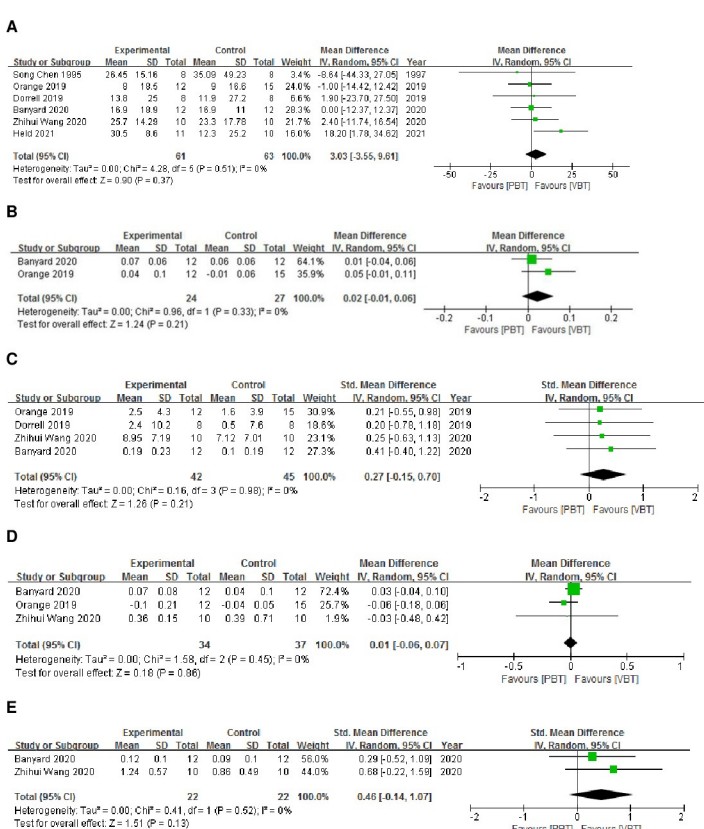

**Fig 3.** Forest plot of the results on strength performance (A), Load velocity 60%1RM (B), jump performance (C), linear sprint performance (D) and change of direction speed performance (E).

**3.4.2 CMJ.** The SMD was calculated due to one study using peak velocity as testing value. The pooled results from 4 studies consisting of 87 participants indicated that there was no significant difference in improving CMJ between VBT and PBT ($p$ = 0.21). The heterogeneity was low (Chi$^2$ = 0.16, $p$ = 0.98; I$^2$ = 0%). In random effects model, aggregated effect size and 95%CI were 0.27 (-0.15, 0.70) classified as a small effect in favor of VBT. These data were showed in Fig 3C.

**3.4.3 Speed.** The pooled results from 3 studies consisting of 71 participants indicated that there was no significant difference in improving speed between VBT and PBT ($p$ = 0.86). The heterogeneity was low (Chi$^2$ = 1.58, $p$ = 0.45; I$^2$ = 0%). In random effects model, aggregated MD and 95%CI were 0.01s (-0.06, 0.07) in favor of VBT. These data were showed in Fig 3D.

**3.4.4 CODs.** The SMD was calculated due to both included studies with different tests. The pooled results from 2 studies consisting of 44 participants indicated that there was no significant difference in improving CODs between VBT and PBT ($p$ = 0.13). In fixed effects model, the heterogeneity was low (Chi$^2$ = 0.41, $p$ = 0.52; I$^2$ = 0%). aggregated effect size and 95%CI were 0.46 (-0.14, 1.07) classified as a small effect in favor of VBT. These data were showed in Fig 3E.

## 4 Discussion

This systematic review with meta-analysis aimed to compare the training effects of VBT vs. PBT on measures of strength, jump, linear sprint and CODs performance. The results showed that both VBT and PBT were effective, with little to no difference between VBT and PBT. However, all pooled effects demonstrated a trend in favor of VBT, which did not support our hypothesis well.

### 4.1 Strength performance

Both VBT and PBT produced similar gains in squat 1RM, with minimal differences presented in mean changes (3.03kg, 95%CI:-3.55, 9.61) in favor of VBT. The result was similar to a recently meta-analysis by Davis who reported that maximal and moderate velocity exhibited similar improvement in muscular strength. However, the effect (ES = 0.31; 95%CI:-0.01, 0.63) was small in favor of higher velocity with the same intensity ranging 60%-79%1RM [31]. The squat is one of the exercises most frequently adopted for enhancing and evaluating physical performance and health due to its similarities of biomechanics to a various of key sports and daily tasks [32, 33]. Interestingly, all included studies of this systematic review applied squat as the main intervention exercise. It is clear that a change in the muscle cross sectional area (CSA) and an enhanced neural adaptation are the two key factors for the improvement of maximum strength [16, 34]. However, to date, there has been a paucity of study conducted to compare the difference on improving muscle mass between VBT and PBT, excepted one study by Fernandez Ortega et al. which demostrated that both VBT and PBT could significantly increase children's (13.6±1.2 years) muscle CSA, but without group difference [21]. Of note, with participants aged under 14 yrs, caution must be applied, as the finding might be limited by the growth effect. Thus, it is still not clear which method is more effective in improving muscle CSA in adults. However, Orange et al. and Banyard et al. both reported that VBT performed significant higher sessional mean velociy and mean power compared to the PBT (ES = 1.25) with the same verbal encouragement for participants' maximal voluntary efforts [18, 22]. the results were likely the consequences of subtle decreases in load [19], and placebo effect from the velocity testing device in VBT group. Although the researches conducted by Chen et al. and Held et. al. did not provide the repetition velocity data in sessions [20, 29], it could be expected that participants with feedback of repetition velocity in sessions might perform even higher repetition velocity than PBT [12]. Furthermore,

interstingly, 4 of the 6 included studies reported that VBT had lower training volume contrasting to PBT with load range 59% to 95%1RM [17, 18, 20, 22]. For example, Held et al. reported that the total repetitions of VBT group were only 77%% of PBT group [17]. Similarly, of 3 studies reported VBT produced lower training stress [17–19], this is likely explained by the reason that the mean time under tension of VBT was significant less than PBT, which might lead the reduction of mechanical stress correspondingly to the practitioners [18, 19]. It is plausible that the lower volume and perceived training stress also contributed partly to the higher repetition velocity owing to participants' less fatigue and need for recovery. Held et al. demonstrated that the indicators of overall recovery and stress after 24 and 48 hours in VBT were superior to PBT via a validated daily questionnaire [17]. These two factors were also contributed to the higher repetition velocity. Due to these differences, the muscle fast fibres are likely to be selectively recruited more when performing faster repetition with the similar load by maximum voluntary efforts. Taken together, it might be inferred that VBT and PBT enhanced strength through different mechanism. VBT should produced a better stimulus for neural adaptations owing to higher repetition velocity and lower training stress, while it was likely for PBT to produce better adaptation in muscle morphology due to higher mechanical and metabolic stress. These influencing factors might be mutually offsetting to result in the similar effects on enhancing squat 1RM between VBT and PBT.

In the past decades, a growing body of studies established that completing more repetition or training to muscle failure with heavy load training might not be essential for greater improvements in muscle strength and hypertrophy in comparison with lower training volumes [8, 35–39]. Conversely, training to muscle failure might even reduce myosin heavy chain IIX percentage [40]. Recently, Banyard et al. reported that there were little to no difference in the average training load between VBT (69.2±7.2%1RM) and PBT (70.9±7.4%1RM) in 6 weeks of back squat training in a daily undulating model [22]. The similar result was also found by Held et al [17]. Furthermore, evidences indicated that exercise repetitions executed at maximum intended velocity could result in greater improvements in 1RM and power in comparison with the submaximal owing to enhancing motor unit firing rate and stimulating the highest threshold type II fiber which have a greater relative hypertrophy than type I fiber [30, 41–43]. Collectively, regardless of the difference in volume, almost similar load and higher velocity specific stimulation in neural muscular adaptation for above mentioned factors could be the major reason, if not the only one, causing the results on strength in favor of VBT compared to PBT.

The results showed that the MD (0.02m/s) of individual squat LVP 60%1RM was negligible between VBT and PBT, which was even less than the measurement error of the velocity testing device [44]. This finding supported that LVP would keep stable against the same relative load after short term training. Gonzalez-Badillo et al. indicated that the individual LVP would remain stable (MD = 0.01m/s) in pre and post-interventioin test irrespective of a mean 1RM improvement of 9.3% in 6 weeks strength training [9]. Although one of the included studies by Banyard et al. indicated that both groups can improve the barbell mean velocity against the absolute load after 6 weeks training [22]. Considering the methodology involved the use of post minus pre intervention value in our meta-analysis, it could be speculated that difference would be close to zero owing to the reliable individual LVP against the same %1RM in short term training. Of note, the effect was only pooled by two studies from traditional resistance model, more researches from different training model such as power training are needed to explore the difference on improving both absolute and relative load velocity between VBT and PBT.

## 4.2 Jump performance

The results demonstrated that VBT was superior to PBT in CMJ with no significant difference. Previous studies indicated that motor unit recruitment and discharge rate, and muscle fiber

type composition were the determinants for the rate of force development [45] and efficacy of muscle-tendon stretch-shortening cycle (SSC), which were considered as the main factors for improving CMJ performance. As explained earlier, VBT was characterized with a higher velocity, lower volume and training stress in each session. Consequently, an enhancement in neural drive to agonist and SSC efficacy could be expected [37, 38]. Comparatively, training to muscle failure in PBT may induce undue fatigue and generate sub-optimal training stimulus, which may lead the adaptations towards slower and endurance resistant fiber types, and impair the rate of force development [38]. Furthermore, Banyard et al. reported that the mean deviation of sessional repetition velocity was greater for the PBT (-13.6 ± 6.8%) contrasted to VBT (-0.2 ± 5.2%) [22]. There is also evidence that the greater consistency of high repetition velocity was superior enhancements in rapid actions [46]. Cumulatively, it should be expected that VBT with lower variability in sessional repetition might have better training effect. Besides, the squat as the main intervention exercise was similar in biomechanics to the CMJ. Although, it is hard to perform sets of repetitions at fast velocity in heavy load as CMJ. Under the velocity specificity of resistance training [47], the higher movement velocity with the maximum voluntary efforts in VBT was likely to result in better transference to CMJ performance compared to PBT. Taken together, it could be speculated that VBT might produce superior change in jump performance compared to PBT.

## 4.3 Linear sprint and CODs performance

The MD in linear sprint was negligible (0.01s) between VBT and PBT. Previous evidence showed that the squat maximum strength existed very large correlation (r = -0.77; p = 0.001) with the sprint performance [48]. Thus, it could be inferred that the similar improvement of strength between VBT and PBT as above mentioned may be the reason for this negligible result. Furthermore, in line with the concept of training specificity, the intervention program without linear sprint training and horizon oriented resistance exercises may lead the lack of specificity transference to the tests measures, which in combination with the relatively short training duration may also impair resistance training adaptations to sprint performance measures [49]. Besides, one included research by Orange et al. found that sprint performance was impaired in both groups. Under the author's explaination, it may be attributed to the accumulated match fatigue in competitive season [18]. This negative effect may also impact the pooled results methodologically. Overall, the similar change of stength, lack of specificity transference and methodological difference of included studies may be the main factors for these negligible results. Of note, there was a paucity of research carried out to compare the different effects on enhancing linear sprint between VBT and PBT, more researches are still needed to distinguish the difference.

The result was in favor of VBT on enhancing CODs with a small effect. However, there was insufficient number of studies to pool the magnitude of effects on CODs in VBT vs. PBT. Of the 2 studies that compared changes in this outcome measure, Banyard et al. indicated that the effect in CODs (ES = 0.67–0.79) favored VBT compared to PBT [22]. Similarly, Wang et al. demonstrated that CODs was the only measure that VBT was superior to PBT [28]. The possible explaination for this result may be the same as above mentioned such as the higher and more consistent repetition velocity. Whether the pooled effect would be enlarged in favor of VBT via more search remains to be determined in future.

From a practical view, irrespective of achieving the similar effects as PBT in enhancing maximum strength, CMJ, linear sprint and CODs performance, VBT had significant lower volume and training stress, and better recovery. This implies that pactitioners are likely more enjoyable with VBT, which might also affect long-term adherence. Furthermore, as mentioned in introduction part, VBT has several compelling features such as estimating the load by

highest repetition velocity in real time, controlling the level of fatigue by velocity loss in sets and augmenting motivation by feedback of repetition velocity. Besides, Hopkins et al. stated the smallest worthwhile enhancement by 10% will help the athletes to win the game [50]. Thus, if the facilities are available, it may be expected that VBT could be preferred over PBT.

## 4.4 Limitations

We acknowledge that there are some limitations in this study. One major drawback is the small number of studies included in meta-analysis, which may prevent us from meta-regression and moderator ananlysis. It is hoped that this limitation will be diluted with more studies conducted in the future. Another limitation is the age of included participants ranging from 16 to 30 yrs. Therefore, our findings are difficult to be extrapolated to all practitioners beyond this range of age. However, our intention was to explore the knowledge on the topic with particular emphasis on the athletic population. Due to one quasi-control study, all results were graded as low quality in accordance with GRADE. Futher rigorously designed randomized control studies are warranted to provide robust evidence. Besides, we only searched the studies written in English and Chinese, thus, some relevent articles in other language would be missed.

## 5. Conclusion

In conclusion, both VBT and PBT were effective in enhancing strength, jump, linear sprint and CODs performance. Although there was no signifincat difference between VBT and PBT upon improving each outcome measures, VBT exhibited lower volume and training stress than PBT. VBT could be more suitable when athletes are suffering from busy training and competition schedule, and for those who would like to focus on enhancing power. In addition, given the less exhaustive nature of VBT, this training may help to mitigate the interference of concurrent endurance and strength training on training adaptations.

## Supporting information

**S1 Dataset.**
(XLSX)

**S1 Table.**
(DOCX)

**S2 Table.**
(DOCX)

**S1 Checklist.**
(DOCX)

## Acknowledgments

The authors thank Samuel T. Orange's advice on methodology.

## Author Contributions

**Conceptualization:** Kai-Fang Liao.

**Data curation:** Kai-Fang Liao, Xin-Xin Wang, Meng-Yuan Han, Lin-Long Li.

**Formal analysis:** Kai-Fang Liao.

**Funding acquisition:** Yong-Ming Li.

**Methodology:** Kai-Fang Liao, Xin-Xin Wang, Meng-Yuan Han, Lin-Long Li, George P. Nassis, Yong-Ming Li.

**Project administration:** Kai-Fang Liao.

**Supervision:** George P. Nassis, Yong-Ming Li.

**Writing – original draft:** Kai-Fang Liao, Xin-Xin Wang.

**Writing – review & editing:** Kai-Fang Liao, George P. Nassis, Yong-Ming Li.

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
