## [Decision Letter · Decision Letter 0]

15 Jun 2021

PONE-D-21-12316

Effects of Velocity Based Training vs. Traditional 1RM Percentage-based Training on Improving Strength, Jump, Linear and Change of Direction Speed Performance: A Systematic Review with Meta-analysis

PLOS ONE

Dear Dr. Li,

Thank you for submitting your manuscript to PLOS ONE. After careful consideration, we feel that it has merit but does not fully meet PLOS ONE’s publication criteria as it currently stands. Therefore, we invite you to submit a revised version of the manuscript that addresses the points raised during the review process.

I was waiting for a second reviewer who finally was not able to send the review report. Therefore, following your request, I am sending you the revisions of one reviewer while clarifying to you that another reviewer will be invited in the second review of revisions.

We look forward to receiving your revised manuscript.

Kind regards,

Daniel Boullosa

Academic Editor

PLOS ONE

Journal Requirements:

2.  Thank you for stating the following in the Tiltle page of your manuscript:

"This work was supported by the Winter Olympics Foundation [2018FF0300901];

China Institute of Sport Science Basic Foundation [Basic 17-30]."

"The funders had no role in study design, data collection and analysis, decision to

publish, or preparation of the manuscript"

Reviewers' comments:

Reviewer's Responses to Questions

**Comments to the Author**

1. Is the manuscript technically sound, and do the data support the conclusions?

Reviewer #1: Yes

2. Has the statistical analysis been performed appropriately and rigorously? 

Reviewer #1: Yes

3. Have the authors made all data underlying the findings in their manuscript fully available?

Reviewer #1: Yes

4. Is the manuscript presented in an intelligible fashion and written in standard English?

Reviewer #1: Yes

5. Review Comments to the Author

Reviewer #1: First of all, thank you for the opportunity to review this well-written meta-analysis. The methods are clearly presented and the results are also interpreted and discussed accordingly. Nevertheless, I have the following specific comments:

Site and Line numbers would simplify the review process considerably.

Abstract:

“However, VBT can achieve a similar training effect with lower training volume and stress.” → Even though I agree with this statement, this is not a result of this meta analysis

Keywords: I would recommend using keywords that are not already used in the abstract. This may increase the reach of the paper.

Introduction:

“VBT was a resistance training intervention that uses velocity feedback to prescribe and/or manipulate training load based on the perfect inversely linear relationship between load and repetition velocity, and nice correlation between velocity loss and maximum repetitions, as well as the %maximum repetitions and level of fatigue in sets,two new variables were adopted for prescribingthe training load in VBT, one is the initial fastest repetition velocity in sets to set the load instead of%1RM, the other is the velocity loss threshold (VL) to terminate the set instead of the traditional fixed repetitions sets.” → This sentence is way to long. Please rewrite this.

“More recently, some scattered researches have been carried out on comparing the training effects of VBT vs. PBT.” → A reference is needed for this sentence.

“One possible explanation is that the effects of the resistance training intervention on physical performance could be affected by several participants’ characteristics such as the initial training status or chorological age, while the other is the differences in resistance training variables such as the exercises, duration, periods, intensity and volume. Specifically, some studies adopted the fixed repetitions schemes to terminate the set in VBT group, whereas others used fixed velocity loss value either separately or in combination with the former.” → I would expect such explanations in the discussion section.

Methods:

The representation of the search strategy is somewhat confusing. why, for example, was "endurance" searched for? A tabular representation might be easier to read and understand.

“Quality of evidence” Is this not (at least partially) redundant to the Pedro score?

“Statistical analysis” → I would mention CHI^2 and I^2 only after the Ramdom effects model has been described.

The formula for SMD is wrong (or is displayed incorrectly).

Results:

were the reference lists of the resulting 6 studies checked for potential additional studies?

please avoid redundant information between text and tables/figures. Also there are some irrelevant informations like (“The MD was calculated.”)

“LVP with 60%1RM was chosen because 1m/s (60%1RM) was usually used to evaluate the performance in resistance training. “ → This sentence needs a reference.

Discussion:

Recheck spelling & grammar

Table 3 is not necessary. In comparison to table 2, no additional information is provided here.

The symmetry of the funnel plot could be additionally checked by egger's p Value.

6. PLOS authors have the option to publish the peer review history of their article (what does this mean?). If published, this will include your full peer review and any attached files.

Reviewer #1: **Yes: **Steffen Held

---

## [Author Response · Author response to Decision Letter 0]

17 Jul 2021

July 7th 2021

Dear Professor Daniel Boullosa

thank you for your letter and the comments from the reviewers with regards to our paper submitted to PLOSONE.

We have carefully checked the manuscript and revised it under the reviewer’s comments. With this letter we submit the modified version as well as the list of changes we have made.

If you have any question, please let me know.

Thank you once again for your feedback.

Sincerely,

Professor Yongming Li

Response to editor’s comments:

Thanks for your comments on our paper. We have revised our paper according to your comments.

Journal Requirements:

Response: we have corrected it based on those requirements. Thank you.

2. Thank you for stating the following in the Title page of your manuscript:

"This work was supported by the Winter Olympics Foundation [2018FF0300901];

China Institute of Sport Science Basic Foundation [Basic 17-30]."

"The funders had no role in study design, data collection and analysis, decision to

publish, or preparation of the manuscript"

Response: We have already deleted it in title page, and put it into our cover letter. Thank you.

Please find below the new statement

"This work was supported by the Winter Olympics Foundation [2018FF0300901];

China Institute of Sport Science Basic Foundation [Basic 17-30] funds to Yongming Li."

Response to reviewer #1:

Thanks for your comments on our paper. We have revised our paper according to your comments.

Reviewer #1: First of all, thank you for the opportunity to review this well-written meta-analysis. The methods are clearly presented and the results are also interpreted and discussed accordingly. Nevertheless, I have the following specific comments:

Site and Line numbers would simplify the review process considerably.

Abstract:

1 “However, VBT can achieve a similar training effect with lower training volume and stress.” → Even though I agree with this statement, this is not a result of this meta analysis

Response: We agree with this, and we have deleted this statement from the conclusion. Thank you.

2 Keywords: I would recommend using keywords that are not already used in the abstract. This may increase the reach of the paper.

Response: We agree with it and we have replaced the keywords. The key words that we are now using are: load, velocity loss, athletic performance, resistance training, muscle strength.

Introduction:

1 “VBT was a resistance training intervention that uses velocity feedback to prescribe and/or manipulate training load based on the perfect inversely linear relationship between load and repetition velocity, and nice correlation between velocity loss and maximum repetitions, as well as the %maximum repetitions and level of fatigue in sets, two new variables were adopted for prescribing the training load in VBT, one is the initial fastest repetition velocity in sets to set the load instead of%1RM, the other is the velocity loss threshold (VL) to terminate the set instead of the traditional fixed repetitions sets.” → This sentence is way to long. Please rewrite this.

Response: We have re-written this section and we have made it shorter. Thank you.

2 “More recently, some scattered researches have been carried out on comparing the training effects of VBT vs. PBT.” → A reference is needed for this sentence.

Response: The references have been added. They are as following:

(1) Held, S., et al., Improved Strength and Recovery After Velocity-Based Training: A Randomized Controlled Trial. Int J Sports Physiol Perform, 2021: p. 1-9. Orange, S.T., et al., Effects of In-Season Velocity- Versus Percentage-Based Training in Academy Rugby League Players. Int J Sports Physiol Perform, 2019: p. 1-8.

(2) Banyard, H.G., et al., Comparison of the Effects of Velocity-Based Training Methods and Traditional 1RM-Percent-Based Training Prescription on Acute Kinetic and Kinematic Variables. Int J Sports Physiol Perform, 2019. 14(2): p. 246-255.

(3) Dorrell, H.F., M.F. Smith, and T.I. Gee, Comparison of Velocity-Based and Traditional Percentage-Based Loading Methods on Maximal Strength and Power Adaptations. J Strength Cond Res, 2020. 34(1): p. 46-53.

(4) Ortega, J.A.F., Y.G.D.l. Reyes, and F.R.G. Pen˜a, Effects of strength training based on velocity versus traditional training on muscle mass, neuromuscular activation, and indicators of maximal power and strength in girls soccer players. Apunts Sports Med, 2020. 55(206): p. 53-61.

3 “One possible explanation is that the effects of the resistance training intervention on physical performance could be affected by several participants’ characteristics such as the initial training status or chorological age, while the other is the differences in resistance training variables such as the exercises, duration, periods, intensity and volume. Specifically, some studies adopted the fixed repetitions schemes to terminate the set in VBT group, whereas others used fixed velocity loss value either separately or in combination with the former.” → I would expect such explanations in the discussion section.

Response: We agree with it. We have deleted this section from the introduction.

Methods:

1 The representation of the search strategy is somewhat confusing. why, for example, was "endurance" searched for? A tabular representation might be easier to read and understand.

Response: We revised the searching strategy, and added a tabular (supplemental table 1) to detail the searching strategy and syntax. Thank you.

2 “Quality of evidence” Is this not (at least partially) redundant to the Pedro score?

Response: Actually, this part is following the latest PRISMA(2019) requirements. The Pedro score we used is to evaluate single included study, but GRADE is for evaluating the overall quality of all included studies. Thank you.

3 “Statistical analysis” → I would mention CHI^2 and I^2 only after the Random effects model has been described.

Response: We adjusted the order of related sentences in order to make it clear that we only used the random effect model to aggregate the effects. The reason for choosing random effect model is because fixed effect model is based on the assumption that the results apply only to a given group of studies, and assumes that variation in effect sizes among studies is due to within-study variance. Random-effects model apply more generally and assumes the true effects from different studies also differ from one another, representing a random sample of a population. Our study aim was to compare the training effect of VBT and PBT, and expect to apply the results more generally, not being limited in within-study variance. 

4 The formula for SMD is wrong (or is displayed incorrectly).

Response: We think it is displayed incorrectly. The formula for SMD is [MDVBT– MDPBT/pooled SDpre].and we have also rewritten the description part to make it clearer. Thank you.

Results:

1 were the reference lists of the resulting 6 studies checked for potential additional studies?

Response: Yes, we checked it.

2 please avoid redundant information between text and tables/figures. Also there are some irrelevant information like (“The MD was calculated.”)

Response: Following a carefully check, we have already deleted some redundant information. Thank you.

3 “LVP with 60%1RM was chosen because 1m/s (60%1RM) was usually used to evaluate the performance in resistance training. “ → This sentence needs a reference.

Response: We have included 2 references for this statement. These are :Banyard, H.G., et al., Superior Changes in Jump, Sprint, and Change-of-Direction Performance but Not Maximal Strength Following 6 Weeks of Velocity-Based Training Compared With 1-Repetition-Maximum Percentage-Based Training. Int J Sports Physiol Perform, 2020: p. 1-11; Pareja-Blanco, F., et al., Effect of movement velocity during resistance training on neuromuscular performance. Int J Sports Med, 2014. 35(11): p. 916-24. We have added the references in the revised manuscript.

Discussion:

1 Recheck spelling & grammar

Response: We have done this in this version too. Thank you.

2 Table 3 is not necessary. In comparison to table 2, no additional information is provided here.

Response: this is also following the PRISMA’s requirements. Table 3 is to provide the detailed information.

3 The symmetry of the funnel plot could be additionally checked by egger's p Value.

Response: We have already added the egger’s test and the p value in the revised manuscript. Thank you.

---

## [Decision Letter · Decision Letter 1]

10 Aug 2021

PONE-D-21-12316R1

Effects of Velocity Based Training vs. Traditional 1RM Percentage-based Training on Improving Strength, Jump, Linear and Change of Direction Speed Performance: A Systematic Review with Meta-analysis

PLOS ONE

Dear Dr. Li,

Thank you for submitting your manuscript to PLOS ONE. After careful consideration, we feel that it has merit but does not fully meet PLOS ONE’s publication criteria as it currently stands. Therefore, we invite you to submit a revised version of the manuscript that addresses the points raised during the review process.

We look forward to receiving your revised manuscript.

Kind regards,

Daniel Boullosa

Academic Editor

PLOS ONE

Reviewers' comments:

Reviewer's Responses to Questions

**Comments to the Author**

1. If the authors have adequately addressed your comments raised in a previous round of review and you feel that this manuscript is now acceptable for publication, you may indicate that here to bypass the “Comments to the Author” section, enter your conflict of interest statement in the “Confidential to Editor” section, and submit your "Accept" recommendation.

Reviewer #2: (No Response)

2. Is the manuscript technically sound, and do the data support the conclusions?

Reviewer #2: Yes

3. Has the statistical analysis been performed appropriately and rigorously? 

Reviewer #2: I Don't Know

4. Have the authors made all data underlying the findings in their manuscript fully available?

Reviewer #2: Yes

5. Is the manuscript presented in an intelligible fashion and written in standard English?

Reviewer #2: No

6. Review Comments to the Author

Reviewer #2: This is a valuable study on an interesting topic in the sports sciences field. However, there are some issues that I feel need to be addressed to bring the study up to publishable standards. Mainly, writing should be reviewed by a professional proofreader or a native-speaker, specially the Discussion section. Moreover, some changes are required throughout the this section in order to provide a deeper explanation of your findings. Number lines would help review process. Please see my specific comments below:

Abstract:

I would change “linear speed” to “linear sprint”. Linear speed is already performance. It would be like jump height. Therefore, it would be better to use: linear sprint throughout the manuscript.

Abstract. Methods. SD has not been previously defined

Abstract. Results. Why sometimes you show MD and others SMD? I would use only SMD. Please see my specific comments below.

Introduction

Third paragraph. These research to date… should read: these researches…

Therefore, A systematic review with… should read: Therefore, a systematic…

Methods

Search strategy.

syntax was shown in supplemental table 1… should read: syntax are shown…

Risk of bias assessment within individual studies

Two authors (The second… should read: the second

Brughelli et al [23]…should read: al.

Statistical analysis

Since MD can be highly influenced by sample characteristics. Would not it be better using always SMD?

SMD can amend some of the problems shown by MD, since it includes SD in the calculations.

Both algorithms of MD and SMD in speed and COD outcomes was adjusted…should read: were

Results

Search results

At last, 6 studies included in the qualitative…should read: were included

Risk of bias assessment

All of studies got high scores… should read: All studies or all the studies

Studies’ Characteristics

characteristics of 6 studies included in qualitative analysis was…should read: were

Among which, Only 1 study…should read: only

Quantitative analysis

load velocity with 40%1RM. Shouldn’t be 60% 1RM?

Discussion

IMHO, I miss you highlight the methodological differences between both approaches throughout the Discussion, especially in the Practical Applications section. Thus, you may help readers to better choose the proper method.

Strength performance

improvement in musclular strength…should read: muscular

similarities of biomechenics to a various…should read: biomechanics

PBT ( ES = 1.25) With the…should read: with

Please rewrite this sentence: This likely accounted for subtle decreases in load adjusted in accordence with targeted velocity

velocity testing device. although the…should read: device, although

…. and placebo effect from the velocity testing device. although the research conducted by

Chen et al. and Held et. al. did not provide the velocity data of training repetitions [20,

29], it could be inferred that participants might perform even higher repetition velocity

due to enhancing motivation and competitiveness from feedback of velocity[12]. Please split or rewrite this sentence

Furthermore, Interstingly,…should read: interestingly

and 48 hours in VBT was superior…should read: were

VBT should produced a better…should read: would produce

while it was likely for PBT to…should read: it is likely

With respect to the effects in faovr of VBT. This sentence should go on.

more repetition…should read: more repetitions

at maiximum intended velocity…should read: maximum

threshold type Ⅱ fiber which have a greater relative hypertrophy than type Ⅱ fiber…should be: type II and type I, respectively

for above metioned factors…should read: mentioned

device[44]. this finding…should read: This

Considering the methodology involved the use of post minus pre intervention value in our meta-analysis, it could. Please use comma when necessary

4.2 Jump performance

and muslce fiber type composition …should read: muscle

development [45] and muscle-tendo …should read: muscle-tendon

Furthermore, Banyard et al. reported that the mean deviation of

sessional repetition velocity was greater for the PBT (-13.6 ± 6.8%) contrasted to VBT

(-0.2 ± 5.2%) [22]. You should explain why these authors reported this and how can influence jump height adaptations.

is also evdence: evidence

intervetion exercise: intervention

resistance training [47], The higher movement… should read: the

Linear Speed and CODs performance

Evidence showed that the squat strength and sprint performance existed very large significant

correlation ( r= -0.77; p = 0.001) [48]. This sentence is hard to follow

above metioned: mentioned

may also impaire resistance: impair

in both group… should read: in both groups [18].

This negetive effects may: negative

This negligible results: these

still needed to distingush the difference: distinguish

CODs (ES = 0.67-0.79) favorited VBT compared to PBT [22]: favored

for this results: these

may be the same as above metioned such: mentioned

recovery. this implies that pactitioners are… shoud read: This implies that practitioners…

4.4 Limitations

all results was graded: were

Conclusion

less excaustive nature: exhaustive

TABLES

Table 2.

Song Chen 1995. I cannot access to this reference. But did they use as a reference load? the maximal power load? or all of them trained at the same velocity? It is strange that they compared PBT and VBT in 1997 when the first publications about using VBT instead of PBT are from 2010.

I guess they compared power-based training, but this is not the same idea, since the use power, not velocity, for monitoring and programing the training load.

Some of the abbreviations described in the footnote are not used in the table.

Table 3.

I miss some of the included studies here

FIGURES

Figure 3A. I guess this should be Song;, C. and M. qiwei, Effects of fixed velocity training methods on load

velocity profile. Journal of Beijing University of Physical Education, 1995. 18(3):

p. 81-88.

Figure 3D. how have you calculated the weight in Fig. 3D? It is weird that a study with 10 subjects means the 1.9% and other with 12 the 72.4% and the third one with 15 subjects and only the 25.7% of the total weight.

7. PLOS authors have the option to publish the peer review history of their article (what does this mean?). If published, this will include your full peer review and any attached files.

Reviewer #2: No

---

## [Author Response · Author response to Decision Letter 1]

29 Sep 2021

Dear Professor Daniel Boullosa

Many thanks for your letter and the comments from the reviewers regarding our paper.

After carefully checking the manuscript, we have revised it according to the reviewer’s comments. We submit here the modified version as well as a list of change.

If you have any question, please let me know.

Thank you again.

Kind regards

Response to editor’s comments:

Thanks for your great comments on our paper. We have revised our paper according to your comments and make sure that the manuscript will be up to publishable level with your help.

Abstract:

1. I would change “linear speed” to “linear sprint”. Linear speed is already performance. It would be like jump height. Therefore, it would be better to use: linear sprint throughout the manuscript.

Response: We agree with your suggestions, and have already changed “linear speed” into “linear sprint” throughout the manuscript. Thank you.

2. Abstract. Methods. SD has not been previously defined.

Response: We have now defined SD. Thank you.

3. Abstract. Results. Why sometimes you show MD and others SMD? I would use only SMD. Please see my specific comments below.

Response: From a practical standpoint, we prefer MD in current study. This is because MD is simpler and more meaningful in practice. For example, if we tell the coaches that VBT can increase 1RM of squat by 10kg in 4 weeks, it will be easy to be understood. On the other side, if we tell them that 1RM can improve by 0.4 whole standard deviation it is very confusing. As a support, the American Psychological Association (APA) Task Force on statistical inference also suggests that if the units of measurement are meaningful on a practical level, then a MD is superior to a SMD. Thus, when all studies report the outcome using the same scale or the same tests, we used MD, otherwise we used SMD. that’s the reason why we sometimes show MD and others SMD. Thank you again.

Introduction

5. Third paragraph. These research to date… should read: these researches…

Response: We have corrected that. Thank you.

6. Therefore, A systematic review with… should read: Therefore, a systematic…

Response: We have corrected that too. Thank you.

Methods

Search strategy.

7. syntax was shown in supplemental table 1… should read: syntax are shown…

Response: This has been corrected.

8. Risk of bias assessment within individual studies

Two authors (The second… should read: the second

Response: We have corrected that. Thank you.

9. Brughelli et al [23]…should read: al.

Response: This has been corrected. Thank you.

Statistical analysis

10.Since MD can be highly influenced by sample characteristics. Would not it be better using always SMD? SMD can amend some of the problems shown by MD, since it includes SD in the calculations.

Response: Thank you for your comment. Previously (response to comment 3) we have justified our approach. Thank you. 

11. Both algorithms of MD and SMD in speed and COD outcomes was adjusted…should read: were

Response: This has been corrected. Thank you. 

Results

Search results

12. At last, 6 studies included in the qualitative…should read: were included

Response: This has been corrected. Thank you. 

Risk of bias assessment

13. All of studies got high scores… should read: All studies or all the studies

Response: This has been corrected. Thank you.

Studies’ Characteristics

14. characteristics of 6 studies included in qualitative analysis was…should read: were

Response: This has been corrected. Thank you.

15. Among which, Only 1 study…should read: only

Response: This has been corrected. Thank you. 

Quantitative analysis.

16. load velocity with 40%1RM. Shouldn’t be 60% 1RM?

Response: Thank you for this comment too. We have corrected that.

Discussion

17. IMHO, I miss you highlight the methodological differences between both approaches throughout the Discussion, especially in the Practical Applications section. Thus, you may help readers to better choose the proper method.

Response: We have added the methodological differences between VBT and PBT in the discussion and practical applications section following your suggestions. Thank you .

Strength performance

18. improvement in musclular strength…should read: muscular

Response: This has been corrected. Thank you. 

19. similarities of biomechenics to a various…should read: biomechanics

Response: We corrected that. Thank you.

20. PBT ( ES = 1.25) With the…should read: with

Response: This has been corrected. Thank you. 

21. Please rewrite this sentence: This likely accounted for subtle decreases in load adjusted in accordance with targeted velocity.

Response: We have rewritten this sentence. Thank you.

22. velocity testing device. although the…should read: device, although

Response: We corrected that too. Thank you.

23. …. and placebo effect from the velocity testing device. although the research conducted by Chen et al. and Held et. al. did not provide the velocity data of training repetitions [20,29], it could be inferred that participants might perform even higher repetition velocity due to enhancing motivation and competitiveness from feedback of velocity[12]. Please split or rewrite this sentence.

Response: We have rewritten this sentences.

24. Furthermore, Interestingly,…should read: interestingly

Response: We have corrected that. Thank you.

25. and 48 hours in VBT was superior…should read: were

Response: This has been corrected. Thank you. 

26. VBT should produced a better…should read: would produce

Response: This has been corrected. Thank you. 

27. while it was likely for PBT to…should read: it is likely

Response: This has been corrected. Thank you. 

28. With respect to the effects in favor of VBT. This sentence should go on.

Response: After considerable thinking, we decided to delete this incompletely sentence. 

29. more repetition…should read: more repetitions

Response: This has been corrected. Thank you.

30. at maiximum intended velocity…should read: maximum

Response: This has been corrected. Thank you.

 31. threshold type Ⅱ fiber which have a greater relative hypertrophy than type Ⅱ fiber…should be: type II and type I, respectively

Response: This has been corrected. Thank you.

 32. for above metioned factors…should read: mentioned

Response: This has been corrected. Thank you.

 33. device[44]. this finding…should read: This

Response: This has been corrected. Thank you.

34. Considering the methodology involved the use of post minus pre intervention value in our meta-analysis, it could. Please use comma when necessary

Response: We have used comma. Thank you.

4.2 Jump performance

35. and muslce fiber type composition …should read: muscle

Response: This has been corrected. Thank you.

36. development [45] and muscle-tendo …should read: muscle-tendon.

Response: This has been corrected. Thank you.

37. Furthermore, Banyard et al. reported that the mean deviation of

sessional repetition velocity was greater for the PBT (-13.6 ± 6.8%) contrasted to VBT(-0.2 ± 5.2%) [22]. You should explain why these authors reported this and how can influence jump height adaptations.

Response: We added a sentence to explain that. Thank you for your comment.

38. is also evdence: evidence

Response: This has been corrected. Thank you.

 39. intervetion exercise: intervention

Response: This has been corrected. Thank you.

 40. resistance training [47], The higher movement… should read: the

Response: This has been corrected. Thank you.

Linear Speed and CODs performance

41. Evidence showed that the squat strength and sprint performance existed very large significant correlation ( r= -0.77; p = 0.001) [48]. This sentence is hard to follow.

Response: We have rewritten this sentence. Thank you.

42. above metioned: mentioned.

Response: This has been corrected. Thank you.

43. may also impaire resistance: impair.

Response: This has been corrected. Thank you.

 44. in both group… should read: in both groups [18].

Response: This has been corrected. Thank you. 

45. This negetive effects may: negative.

Response: This has been corrected. Thank you.

46. This negligible results: these.

Response: This has been corrected. Thank you.

47. still needed to distingush the difference: distinguish.

Response: This has been corrected. Thank you.

48. CODs (ES = 0.67-0.79) favorited VBT compared to PBT [22]: favored.

Response: This has been corrected. Thank you 

49. for this results: these

Response: This has been corrected. Thank you.

 50. may be the same as above metioned such: mentioned

Response: This has been corrected. Thank you.

51. recovery. this implies that pactitioners are… shoud read: This implies that practitioners…

Response: We corrected that too. Thank you.

4.4 Limitations

52. all results was graded: were

Response: We corrected that. Thank you.

Conclusion

53. less excaustive nature: exhaustive

Response: This has been corrected. Thank you.

TABLES

Table 2.

54. Song Chen 1995. I cannot access to this reference. But did they use as a reference load? the maximal power load? or all of them trained at the same velocity? It is strange that they compared PBT and VBT in 1997 when the first publications about using VBT instead of PBT are from 2010.

I guess they compared power-based training, but this is not the same idea, since the use power, not velocity, for monitoring and programing the training load.

Response: After carefully rechecking this reference we have made sure that this study by Song Chen 1995 trained the VBT group at the same velocity by adjusting the load with a self-made velocity testing devices, which is in accordance with our definition of VBT described in the method. As you correctly mentioned, Chen’s concept was associated with power-based training at that time. However, the author puts forward some quite new and unique insight on velocity load such as the range of speed control (velocity loss), increasing the speed with controlled weight, and increasing weight with controlled speed. This same author also published several articles from his doctoral thesis to state the concept of velocity load training and provided suggestions on how to prescribe the load using velocity. Please find below one of his articles’ abstract published in 1994. The full reference for that is Song Chen, Qiwei Ma. Training theory and method of developing velocity-force by quantitative control of movement speed. Journal of Beijing University of Physical Education. 1994; 17(2):78-89. 

55. Some of the abbreviations described in the footnote are not used in the table.

Response: Thank you for this suggestion too. After checking we have deleted the extra abbreviations. 

Table 3.

56. I miss some of the included studies here

Response: You are right, two studies were not in the table 3. Because table 3 is to show the results of dealing with the multiplicity in included studies. Therefore, we did not include the study without multiplicity in table 3. Thank you.

FIGURES

57. Figure 3A. I guess this should be Song;, C. and M. qiwei, Effects of fixed velocity training methods on load velocity profile. Journal of Beijing University of Physical Education, 1995. 18(3):p. 81-88.

Response: Yes, that’s Song Chen and Qiwei MA. We have corrected it in the figure. Thank you.

58. Figure 3D. how have you calculated the weight in Fig. 3D? It is weird that a study with 10 subjects means the 1.9% and other with 12 the 72.4% and the third one with 15 subjects and only the 25.7% of the total weight.

Response: Thank you for this point too. This is because we adopted the random-effects model to calculate the MD in figure 3D via Revman software. Weighting within the random-effects model assumes two sources of variability in effects, one from sampling error and one from study level differences. However the latter is based on the variability in effect sizes across the group of studies and it has been taken to imply that the random-effects model assumes a distribution of true population effects from which the observed studies are sampled. The reason for this large difference in weight distribution may be due to the fact the third one with much bigger SD than the other two studies results in large study level difference.

---

## [Decision Letter · Decision Letter 2]

27 Oct 2021

Effects of Velocity Based Training vs. Traditional 1RM Percentage-based Training on Improving Strength, Jump, Linear Sprint and Change of Direction Speed Performance: A Systematic Review with Meta-analysis

PONE-D-21-12316R2

Dear Dr. Li,

We’re pleased to inform you that your manuscript has been judged scientifically suitable for publication and will be formally accepted for publication once it meets all outstanding technical requirements.

Kind regards,

Daniel Boullosa

Academic Editor

PLOS ONE

Additional Editor Comments (optional):

Reviewers' comments:

Reviewer's Responses to Questions

**Comments to the Author**

1. If the authors have adequately addressed your comments raised in a previous round of review and you feel that this manuscript is now acceptable for publication, you may indicate that here to bypass the “Comments to the Author” section, enter your conflict of interest statement in the “Confidential to Editor” section, and submit your "Accept" recommendation.

Reviewer #2: All comments have been addressed

2. Is the manuscript technically sound, and do the data support the conclusions?

Reviewer #2: Yes

3. Has the statistical analysis been performed appropriately and rigorously? 

Reviewer #2: Yes

4. Have the authors made all data underlying the findings in their manuscript fully available?

Reviewer #2: Yes

5. Is the manuscript presented in an intelligible fashion and written in standard English?

Reviewer #2: Yes

6. Review Comments to the Author

Reviewer #2: (No Response)

7. PLOS authors have the option to publish the peer review history of their article (what does this mean?). If published, this will include your full peer review and any attached files.

Reviewer #2: No

---

## [Editor Report · Acceptance letter]

5 Nov 2021

PONE-D-21-12316R2 

Effects of Velocity Based Training vs. Traditional 1RM Percentage-based Training on Improving Strength, Jump, Linear Sprint and Change of Direction Speed Performance: A Systematic Review with Meta-analysis 

Dear Dr. Li:

I'm pleased to inform you that your manuscript has been deemed suitable for publication in PLOS ONE. Congratulations! Your manuscript is now with our production department. 

Kind regards, 

on behalf of

Dr. Daniel Boullosa 

Academic Editor

PLOS ONE